# Treatment Planning in Intraoperative Radiation Therapy (IORT): Where Should We Go?

**DOI:** 10.3390/cancers14143532

**Published:** 2022-07-20

**Authors:** Carlo Cavedon, Renzo Mazzarotto

**Affiliations:** 1Medical Physics Unit, Azienda Ospedaliera Universitaria Integrata, 37124 Verona, Italy; 2Radiation Oncology Unit, Azienda Ospedaliera Universitaria Integrata, 37124 Verona, Italy; renzo.mazzarotto@aovr.veneto.it

**Keywords:** IORT, TPS, patient safety, in-room imaging, dose calculation

## Abstract

**Simple Summary:**

Intraoperative radiation therapy is evolving towards new treatment regimens, including ultrahigh dose rates, while in-room imaging systems are increasingly being used for treatment planning and verification. Furthermore, the combination of multiple treatment modalities is being investigated and suggested in some studies, with the aim to improve clinical outcomes. This evolution calls for newly designed treatment planning systems in which several features shall be integrated. In this article, an attempt is made to identify emerging needs and foresee the possible evolution of treatment planning technology and strategies for intraoperative radiation therapy in the near future.

**Abstract:**

As opposed to external beam radiation therapy (EBRT), treatment planning systems (TPS) dedicated to intraoperative radiation therapy (IORT) were not subject to radical modifications in the last two decades. However, new treatment regimens such as ultrahigh dose rates and combination with multiple treatment modalities, as well as the prospected availability of dedicated in-room imaging, call for important new features in the next generation of treatment planning systems in IORT. Dosimetric accuracy should be guaranteed by means of advanced dose calculation algorithms, capable of modelling complex scattering phenomena and accounting for the non-tissue equivalent materials used to shape and compensate electron beams. Kilovoltage X-ray based IORT also presents special needs, including the correct description of extremely steep dose gradients and the accurate simulation of applicators. TPSs dedicated to IORT should also allow real-time imaging to be used for treatment adaptation at the time of irradiation. Other features implemented in TPSs should include deformable registration and capability of radiobiological planning, especially if unconventional irradiation schemes are used. Finally, patient safety requires that the multiple features be integrated in a comprehensive system in order to facilitate control of the whole process.

## 1. Introduction

The last two decades have seen unparalleled development in treatment planning for external beam radiation therapy (EBRT) [1]. Evolutions in dose calculation algorithms, optimization strategies and the availability of advanced imaging systems has laid to the capability of designing, evaluating, and verifying high-resolution dose distributions almost “painted” to optimally cover target volumes, while sparing sensitive structures and nearby healthy tissue as much as possible. Accuracy in dose calculation has reached levels that guarantee sufficient confidence when pondering slightly different solutions between rival treatment plans—a task often jeopardized by poor accuracy in sub-optimal treatment planning settings. In clinical practice, limiting factors in EBRT treatment planning are nowadays probably intrafraction motion management and day-by-day variations, that would require the extensive use of adaptive radiotherapy approaches.

The same cannot be said for treatment planning in intraoperative radiation therapy. Treatment planning solutions, in this case, are not characterized by diverse strategies nor are they implemented with the richness of options currently available in EBRT. The difficult modelling of the anatomical district to treat through proper imaging techniques—unlike EBRT, where pre-treatment accurate imaging solutions are available—seriously hinders the capacity of treatment planning systems to produce reliable, predictive and robust dose distributions. Beam shaping devices (applicators) and protective gear such as disk-shaped shields to use downstream the electron beam interfere with imaging devices and pose dosimetric issues due to their high-Z chemical composition [2].

On the other hand, IORT is administered in one single fraction so no inter-fraction modifications occur, thus eliminating the need for inter-fraction treatment adaptation and simplifying the whole process. Intrafraction motion is also radically reduced compared to EBRT: another potential advantage over treatment planning systems designed for fractionated external irradiation.

Treatment planning for kilovoltage X-ray based IORT [3] is relatively simple compared to high-energy electron techniques. However, due to the radically different irradiation geometry and dosimetric characteristics of X-ray IORT systems, several difficulties encountered in electron beam techniques apply to the former type of equipment as well.

The time is probably ripe for a change. Availability of accurate dose calculation algorithms for clinical use, such as real-time Monte Carlo calculation [4,5,6], the prospected coming of radically new irradiation schemes such as FLASH therapy [7], and the possibility to use in-room imaging [8] call for an evolution of treatment planning systems in IORT.

In the following sections, after an analysis of currently available technology, an attempt will be made to focus on the needs and foresee the possible evolution of treatment planning systems and strategies dedicated to intraoperative radiation therapy in the near future.

## 2. Current Technology

Dedicated IORT systems have seen a continuous technological evolution in recent years. However, there has been no dramatic change in the general scheme of operation of IORT equipment, and treatment planning systems could therefore be adapted accordingly.

Nonetheless, dose calculation algorithms in current systems may lack the accuracy that characterizes analogous devices dedicated to EBRT. Monte Carlo dose calculation is currently implemented in treatment planning systems dedicated to electron-beam IORT [4,5,6,9,10]. However, most systems still employ suboptimal, simplified algorithms that cannot guarantee a dosimetric accuracy at the same level of systems developed for advanced photon-beam modulated techniques. The same can be said for optimization strategies implemented in treatment planning systems: the relatively simple and standardized irradiation schemes in IORT did not push manufacturers to implement advanced optimization algorithms. For example, while in EBRT commercially available planning software includes multicriteria optimization and advanced autoplanning [11,12], direct planning is still largely in use in IORT.

In-room imaging systems are not routinely used in intraoperative radiation therapy. Several studies can be found in the literature that propose imaging tools aimed at capturing the real-time anatomical situation and therefore adapt treatment planning online [13], but none of them has reached a wide clinical use yet.

All these aspects (new operational schemes, calculation accuracy, optimization strategies, and adaptation based on in-room imaging) may need a sudden change, for the reasons outlined in the following paragraphs.

## 3. Needs and Opportunities

Matching the dosimetric accuracy of EBRT is probably the most compelling need in modern IORT. Inaccuracy due to the irregular anatomical environment, dynamically evolving e.g., due to fluid filling of cavities, and complex scattering phenomena demand fine modelling, accuracy of dose calculation and fast adaptation [14]. Dose distributions expected to be regular and flat may present hot and cold spots due to the scatter produced by sharp irregularities on the irradiated surface [15].

The availability of in-vivo dosimetry in IORT [16] will probably be subject to further development, also because of regulatory requirements. For example, the adoption in the European Community of the basic safety standards dictated by directive 2013/59 EURATOM has led to national regulations requiring explicit patient-based dosimetric verification in case of non-standard and high-dose procedures. However, the most important reason why in-vivo dosimetry will most likely evolve at a higher level compared to current implementations is its capability of offering the basis for real-time treatment adaptation [17,18]. Of course, this must be paralleled by the development of treatment planning capabilities, both for fast and accurate dose calculation accuracy and for readily available optimization tools.

These challenging points are even exacerbated in case of dose escalation protocols and novel irradiation paradigms, e.g., flash therapy [7,19,20]. Since these treatment schemes are often used within controlled clinical studies, dosimetric accuracy is of particular importance in view of the clinical information that is expected to emerge from such experimental treatment regimens. Flash therapy is characterized by tissue sparing capabilities whose mechanism of action is not fully known so far. One of the most probable reasons is oxygen depletion occurring in normal tissues due to the extremely high dose-rate in the initial part of treatment [21]. Whatever the mechanism, however, there is a strong need for inclusion of radiobiological modelling in the treatment planning systems. The biological effect of radiation is dependent on multiple and complex factors, e.g., production of chemical reactive species, inflammatory processes and the consequent recruitment of immune cells, and altered acidic conditions throughout the irradiated volume [21]. Mechanisms of repair at the subcellular level—strongly dependent on the dose rate and the chemical environment—contribute heavily on the overall effect of ionizing radiation. It has been shown in a range of studies that genetic damage is the primary mechanism of radiation therapy [22]. Double-strand breaks (DSBs) of the DNA are responsible for tumour cell lethality. Therefore, there is research ongoing attempting to model cross-sections of DNA to use in advanced models (e.g., Monte Carlo) to provide accurate predictions of radiation damage [23,24]. In view of the complexity described above, failing to include relevant information may cause severe errors and potentially result in patient harm. The task is far from simple, however: interplay between factors, patient-dependent response to treatment and concomitant factors that may be yet unnoticed require a complex approach that would need extensive validation before being introduced in the clinical practice.

IORT can be combined to external beam photon irradiation, which might be administered at different time points. In such cases, it is important to have a tool capable of dose accumulation to calculate and visualize the dose distribution emerging from the combination of treatments. This is probably one of the key points in an ideal treatment planning system dedicated to IORT. The possible combination of IORT and EBRT or even brachytherapy [25] in new treatment paradigms—especially those that include ultrahigh dose rates with electron beams—demands dose accumulation capabilities implemented in clinically-available TPSs. This would entail the availability of accurate deformable registration algorithms to map the dose distribution obtained with one treatment modality onto the new anatomical situation (sometimes radically new) found at the time of the second treatment. A direct opportunity to use advanced deformable registration comes from treatment planning systems or treatment management software developed for EBRT [26,27]. No special development would be necessary to translate these systems to the IORT scenario.

Beam modifiers and shields have been used since the very first experiences in IORT. Nonetheless, the prospected use of 3-D printed beam modifiers and compensators [28] offers the opportunity of delivering personalized treatments that might potentially overcome the problems due to irregularity of surfaces and uncontrolled scatter in the treated area. Of course, such devices should be properly modelled by treatment planning software to allow real time adaptation of the treatment. Although 3-D printed objects do not involve high-Z materials in general, their chemical composition might require a non-trivial dosimetric characterization or the use of Monte Carlo dose calculation to allow a sufficient dosimetric accuracy to be obtained.

From the technological standpoint, a most important development would be the availability of devices for in-room real-time imaging. This should be paralleled by appropriate capabilities of the treatment planning system: seamless import of 3D datasets (e.g., CBCT [29]), calculation of the dose distribution expected from the imaged anatomical situation and the consequent real-time adaptation of treatment plans, and possibly tools for in-vivo treatment verification. 3D datasets obtained by tomographic modalities may not be strictly necessary, however: several studies proposed the use of surface matching [30], X-ray projections, EM beacons or other surrogate signals for the task of treatment adaptation. The concept of adapting a treatment plan to the anatomical situation described in real time immediately before delivery is sometimes referred to with the term “intraplanning” [6], a capability that should be standard equipment of the next-generation treatment planning systems for IORT.

X-ray based IORT shares some of the above but has special needs. The extremely steep dose gradient typical of kilovoltage systems (e.g., 50 kV) compared to electron beam IORT requires a very high spatial accuracy. Furthermore, accounting for the actual tissue composition and the material of the applicators normally used with such sources requires high dosimetric accuracy. In contrast, treatment planning systems provided by vendors typically calculate dose distributions in water, without considering any of the heterogeneities present. Deviations from intended dose prescription have been described, up to 34% in the case of breast irradiation and larger than 300% in bone [31,32]. Therefore, dosimetric accuracy is probably the most important need in kilovoltage-based IORT treatment planning. Studies have been reported in the literature that show the improvement achievable by means of Monte Carlo calculation [4], but no advanced dose calculation algorithm has been proposed so far in commercially available treatment planning software.

## 4. Prospected Development and Possible Strategies

Developments will probably come both from the industry and academic research. However, the wide adoption of advanced treatment planning tools would only be possible if manufacturers chose to implement the most recent developments in commercially available systems. Transfer of currently available technology used for EBRT should be a seamless way to offer advanced TPS capacity in IORT.

Implementation of pure Monte Carlo dose calculation algorithms would require higher computational capacity compared to current systems. The industry should be aware that this is a critical aspect for treatment planning and consequently be prepared to offer adequate tools. Stand-alone workstations may be replaced by cloud-based systems, provided that stable network connection is guaranteed where needed.

A most important strategy to raise the quality level of IORT would be the adoption of imaging devices in the operating room, dedicated to monitoring the anatomical situation of the region to irradiate and to acquiring images and data for real-time adaptation of the treatment plan. This includes portable imaging devices as well as dedicated fixed systems [4,33]. Cone beam CT (CBCT) imaging systems might be the most suitable device for intraoperative imaging [13]. They have the potential of providing excellent tomographic imaging from both standpoints of geometric accuracy and dosimetric information (map of attenuation coefficients to use for treatment planning): extensive experience in EBRT shows that translation to the IORT setting should be sufficiently smooth to allow systems equipped with CBCT capability to be rapidly introduced into clinical practice [34]. Availability of in-room imaging systems would be an extremely important aspect to guarantee the necessary level of accuracy for new treatment schemes, such as the use of ultrahigh dose rates.

Interoperability between existing system is necessary, for example to allow dose accumulation estimates in case of multiple treatments that may include EBRT. This task necessarily involves deformable registration, for example to map the dose distribution administered in IORT to the anatomy at the time of a subsequent external beam treatment. Deformable image registration (DIR) shall also be used when pre-surgical imaging is adapted to the situation in the operating room. Using DIR is a complex task that requires skilled operators and robust algorithms: for example, balancing the amount of deformation with the necessary regularization of the underlying spatial transformation is a critical aspect that needs both experience and reliable algorithms to prevent errors. Special attention should be put by manufacturers to the implementation of DIR: solutions already available for EBRT treatment planning systems might be inadequate for the special needs of IORT.

Finally, it is desirable that the foreseen solutions would be implemented into a fully integrated TPS with advanced image processing tools, including reliable DIR. Integration guarantees that the workflow in the operating room does not suffer from weak points that might represent increased factors of risk. The safe use of radiation sources for medical applications requires that every step of the process is clearly defined and effectively controlled: dealing with multiple, non-integrated systems is feasible and sometimes necessary, but is not an optimal strategy as far as patient safety is concerned.

## 5. Conclusions

New treatment regimens such as ultrahigh dose rates and combination with multiple treatment modalities, as well as the increasing availability of dedicated in-room imaging systems, are the factors that will probably shape the next generation of treatment planning systems in IORT. Interoperability between systems is also a key factor that should be guaranteed in newly developed TPSs. Features should include reliable DIR and capability of radiobiological planning, especially if unconventional irradiation schemes are used.

The need for dosimetric accuracy should encourage the implementation of fast yet accurate Monte Carlo dose calculation algorithms, fast enough to be routinely used in the clinical setting.

Finally, patient safety requires that the multiple features be integrated in a comprehensive system in order to facilitate control of the process.

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
