# Peer review of "Treatment Planning in Intraoperative Radiation Therapy (IORT): Where Should We Go?"

_cancers, 2022, doi:10.3390/cancers14143532_

Round 1

Reviewer 1 Report

The existent well written manuscript illustrates as first the current technical stage as well as  a comprehensive and complex foresee of treament-planning tools in intraoperative radiation therapy (IORT).

However , in the following there are only few remarks as suggestions for the authors:

 Minor remarks:

-        Page 3, „needs and opportunities“ third paragraph: „The biological effect of radiation is dependent on multiple and complex factors e.g.  …… []:

I would suggest to quote and shortly elaborate some potentially most relevant factors (incl. some reference to apprpropriate literature)

-        Page 5, „Prospected development and possible strategies“, second paragraph:

I would suggest to add an explanaition for the abbreviation „DIR“  (deformable registration ?)

Author Response

Thank you for your positive comments and for suggestions aimed at further improvement.

Relevant factors that influence the cellular response to radiation have been included at page 3, as suggested. Reference to the literature has been added accordingly. The paragraph has been therefore modified as follows: "... e.g. production of chemical reactive species, inflammatory processes and the consequent recruitment of immune cells, and altered acidic conditions throughout the irradiated volume [21]. Mechanisms of repair at the subcellular level - strongly dependent on the dose rate and the chemical environment – contribute heavily on the overall effect of ionizing radiation. It has been shown in a range of studies that genetic damage is the primary mechanism of radiation therapy [22]. Double-strand breaks (DSBs) of the DNA are responsible for tumour cell lethality. Therefore, there is research ongoing attempting to model cross-sections of DNA to use in advanced models (e.g. Monte Carlo) to provide accurate predictions of radiation damage [23-24]. In view of the complexity described above, failing to include..."  (newly introduced references are 22-24)

The acronym "DIR" has also been introduced as "Deformable image registration" at its first occurrence at page 5, as suggested. 

A revised version of the manuscript with highlighted changes is attached.

Reviewer 2 Report

I suggest to consider to incorporate some references concerning developmental aspectcts mentioned in the paper: 

- Section "needs and opportunities" topic surface  assessement

García-Vázquez V, Sesé-Lucio B, Calvo FA, Vaquero JJ, Desco M, Pascau J. Surface scanning for 3D dose calculation in intraoperative electron radiation therapy. Radiat Oncol. 2018 Dec 7;13(1):243.

- Section "needs and opportunities" topic in vivo dosimetry

López-Tarjuelo J, Morillo-Macías V, Bouché-Babiloni A, Boldó-Roda E, Lozoya-Albacar R, Ferrer-Albiach C. Implementation of an intraoperative electron radiotherapy in vivo dosimetry program. Radiat Oncol. 2016 Mar 15;11:41

- Section "needs and opportunities" topic cone beam CT

García-Vázquez V, Marinetto E, Guerra P, Valdivieso-Casique MF, Calvo FÁ, Alvarado-Vásquez E, Sole CV, Vosburgh KG, Desco M, Pascau J. Assessment of intraoperative 3D imaging alternatives for IOERT dose estimation. Z Med Phys. 2017 Sep;27(3):218-231

- Section "prospected development..." topic operating room

García-Vázquez V, Marinetto E, Santos-Miranda JA, Calvo FA, Desco M, Pascau J. Feasibility of integrating a multi-camera optical tracking system in intra-operative electron radiation therapy scenarios. Phys Med Biol. 2013 Dec 21;58(24):8769-82

Author Response

Thank you for your positive comments and for suggestions aimed at further improvement. In fact, we failed to include the relevant literature suggested by the Reviewer. References have been added in the appropriate sections. 

A revised version of the manuscript with highlighted changes is attached.
